# Multi-Camera Person Re-Identification Based on Trajectory Data

Diogo Mendes [1], Simão Correia [1], Pedro Jorge [1], Tomás Brandão [1,2], Patrícia Arriaga [1,3] and Luís Nunes [1,2,*]

1   ISCTE—University Institute of Lisbon, 1649-026 Lisbon, Portugal; diogo_amaro_mendes@iscte-iul.pt (D.M.);
    simao_correia@iscte-iul.pt (S.C.); pedro_antonio_jorge@iscte-iul.pt (P.J.); tomas.brandao@iscte-iul.pt (T.B.)
2   ISTAR-IUL—Information Sciences, Technologies and Architecture Research Center, 1649-026 Lisbon, Portugal
3   CIS-ISCTE—Center for Psychological Research and Social Intervention, 1649-026 Lisbon, Portugal
*   Correspondence: luis.nunes@iscte-iul.pt

**Abstract:** This study presents a trajectory-based person re-identification algorithm, embedded in a tool to detect and track customers present in a large retail store, in a multi-camera environment. The customer trajectory data are obtained from video surveillance images captured by multiple cameras, and customers are detected and tracked along the frames that compose the videos. Due to the characteristics of a multi-camera environment or the occurrence of occlusions, caused by objects such as shelves or counters, different identifiers are assigned to each person when, in fact, they should be identified with a unique identifier. Thus, the proposed tool tries to solve this problem in a scenario where there are constraints in using images of people due to data privacy concerns. The results show that our method was able to correctly re-identify the customers present in the store with a re-identification rate of 82%.

**Keywords:** person re-identification; trajectory; multi-camera; object detection; object tracking; computer vision





## 1. Introduction

Retail companies have been going through adaptation processes due to the digitalization that has occurred over the last few years, along with increasing consumer demands. As a result, large retailers are increasingly seeking to understand customer behaviors to predict and meet their needs and, consequently, increase sales.

With the increase in the amount of data that can be collected in these stores, such as images from video surveillance cameras, and thanks to scientific advances in the field of computer vision, it is now possible to identify individuals in a series of consecutive video frames. This allows for the extraction of the path these individuals take during their time in the store.

Predicting the trajectory a customer takes and analyzing their behavior using factors such as the time spent in the store, the areas visited, or their walking speed are some of the potential applications derived from these customer trajectory data [1]. For these applications to be effective, it is necessary that the data used are as accurate as possible to the customer's real trajectory. Achieving this can be challenging with just computer vision techniques due to obstructions in the camera's view (e.g., objects), poor lighting conditions, or the complexities of re-identifying an object or individual across multiple cameras.

Several authors tried to address some of these specific concerns. For instance, ref. [2] proposed an algorithm to identify and track targets captured in images across multiple cameras. They employed B-spline polynomials to approximate and define target trajectories, providing a smoother representation of rapidly moving targets. The method then uses homography transformations to compare target trajectories from different cameras, re-identifying corresponding targets based on trajectory similarities. This approach

relies on finding similarities in the trajectories across different cameras to re-identify individuals. Similarly, ref. [3] created a dataset for multi-target multi-camera tracking (MTMCT) derived from the video game GTA V. The dataset, entitled Multi Camera Track Auto (MTA), comprises videos recorded on six different cameras, each with more than 100 min of footage, showcasing more than 2800 bots (individual identities). Beyond creating the dataset, they proposed a system for bot detecting, re-identification, and tracking, further calculating the distances they traveled and associating trajectories. The process of associating trajectories adopts a clustering methodology, grouping bot trajectories based on distance measures. These measures are calculated using weighted aggregation of five individual distances or constraints, encompassing factors such as the presence of tracks on one or more cameras, homography matching distance, linear prediction discounts, and appearance features. The work of [4] is also related. They proposed a method for tracking people in crowded scenes using multiple cameras to address the occlusions. The method focused on the detection of individuals' heads to extract their trajectory. As such, there is no need to identify the entire body to detect the presence of a person in a specific location. However, this approach might face challenges in situations where individuals' faces need to be anonymized for privacy reasons, making the detection more complex.

The authors in [5] introduced a system designed for real-time re-identification of individuals by leveraging trajectory prediction. To predict a person's movement, it uses a method that generates an image trajectory based on visual data and image coordinates. This approach differs from ours in the way it uses image data to make the re-identification.

In another study, ref. [6] proposed a method to detect individuals via surveillance cameras, generating a heatmap depicting the most visited locations within a store. Their detection algorithm employed YOLOv5 complemented by a homography matrix to represent the projection coordinates of individuals inside in the store.

Building on previous works, the goal of our study is to develop a framework capable of extracting customer trajectories through images captured by video surveillance cameras and re-identify the individuals whose identification is lost due to the nature of a multi-camera setup. To extract the trajectory undertaken by customers and track their movements as they navigate the store, we used computer vision algorithms. Distinct from previous work, the primary contribution of our study lies in the creation of a trajectory-based re-identification method that is able to associate the same identifier (ID) with an individual across different cameras, eliminating the need to use image data directly for this purpose.

The structure of this paper is organized as follows: In Section 2, we describe the dataset that was used in the experiments, present the detection and tracking algorithms used to extract the trajectory points, and explain the customer re-identification framework that was developed; Section 3 provides a description and analysis of the achieved results; and Section 4 draws the conclusions and outlines suggestions for future work.

## 2. Materials and Methods

In this section, we outline our methodology in detail. First, we describe the dataset and its acquisition conditions. Subsequently, we explain the object detection and tracking algorithms that allow us to follow and extract the position of each customer within the store over time, thereby deriving their trajectories. This is followed by a description of the necessary data preparation steps to make the data suitable for re-identification. Finally, we present the re-identification process, which serves to reassign the customer IDs generated from different cameras into a common identifier for each customer. The framework pipeline can be visualized in Figure 1.

### 2.1. Dataset Description

In this study, we use a dataset derived from video footage captured by four surveillance cameras situated in a large retail store. The video footage was collected on seven consecutive days, amounting to a total of 91 h. Each video has a 1080 p resolution and was recorded at a frame rate of 20 fps. The videos are presented in an accelerated format

where a single second of footage corresponds to 6 real-time seconds. The video footage in the dataset covers only a specific section of the store. To ensure privacy, all individuals underwent anonymization using a face-blurring algorithm prior to our analysis. Additionally, a homography for each camera was provided in the form of a projection matrix. This matrix facilitates the conversion of trajectory data extracted from the videos to be projected into a 2D floor plan representation of the store. An illustrative example, captured from one of the surveillance cameras, can be viewed in Figure 2.

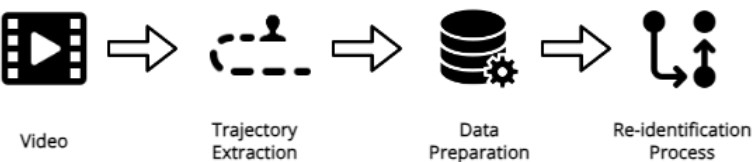

**Figure 1.** Framework pipeline.

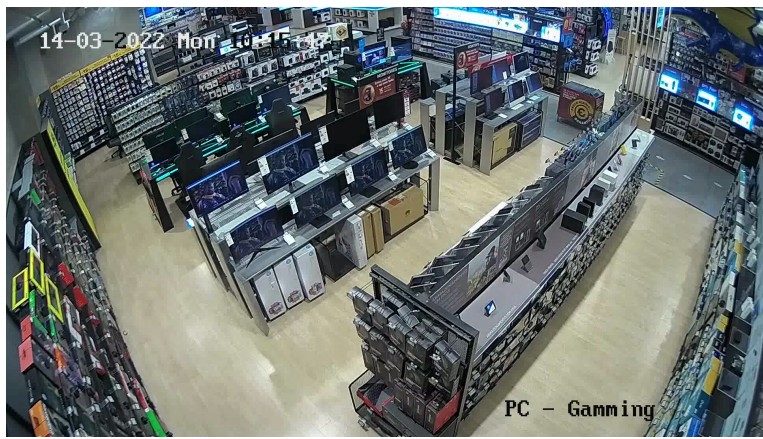

**Figure 2.** Video frame sample of the store area.

### 2.2. Trajectory Extraction

Each video is loaded and decoded into a sequential series of frames, which are stored in a list. For each frame within the list, the object detection algorithm is used to locate the customers. Following the detections generated by YOLOv5, an object tracking algorithm is applied, assigning a unique identifier to each customer as they move over time. Figure 3 shows this initial step of the pipeline encompassing the extraction of the trajectories from the videos.

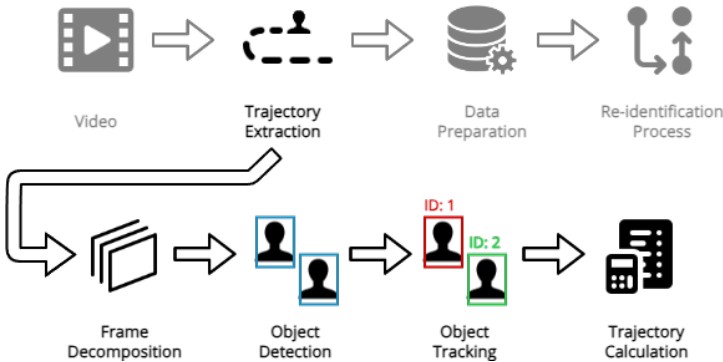

**Figure 3.** Trajectory extraction pipeline.

### 2.2.1. Object Detector

For the detection of customers within our video frames, we used the widely recognized object detector "You Only Look Once" (YOLO), first proposed in [7]. The architecture of this model consists of a single convolutional neural network (CNN), capable of predicting multiple bounding boxes as well as associated class probabilities for each. Its name describes its ability to require just a single analysis of an image to infer the objects within the image and their locations. The main advantages over other object detectors, such as Region-based CNN [8] (including Fast [9] and Faster [10]), are its remarkable speed, its ability to consider the context objects appear in in the image, and its versatile approach to learning object representations. Over time, different variations of the YOLO algorithm have been proposed, the latest being version 8. However, for our work, we decided to use version 5 (YOLOv5), because of its stability, good performance, compatibility with our available computational power, and its ready accessibility through the PyTorch Hub's repository of pre-trained models.

### 2.2.2. Object Tracker

ByteTrack was proposed in [11] and represents the current state of the art in the realm of object tracking algorithms. It adheres to the tracking-by-detection paradigm, which involves using bounding boxes generated with object detection algorithms to make the association of each bounding box over time based on numeric IDs. To take advantage of all detections generated by object detectors, ByteTrack classifies them based on confidence levels (low or high). The decision to include the detections with low confidence levels is grounded in the observation made by the authors that many of these detections often emerge from occlusions. This occurrence does not necessarily compromise or invalidate their utility. Consequently, bounding boxes with confidence surpassing a certain threshold are associated with predictions made with the Kalman Filter [12]. This association is based on either motion (IoU) or appearance (Re-ID) similarity. Then, the Hungarian Method [13] is used to assign the IDs based on the retrieved similarity information. If these associations are unsuccessful, the process is reiterated for detections with confidence below the set threshold through IoU in order to solve occlusions and background detections.

Among object trackers, DeepSORT [14] also stands out for its effectiveness, as highlighted in various studies [15,16]. However, given ByteTrack's superior performance over DeepSORT, we opted for ByteTrack [11] as the tracking algorithm in our research.

### 2.2.3. Trajectory Calculation

Based on the information obtained with the tracking algorithm, we generated a file detailing the bounding box coordinates for every individual detected in the videos. Additionally, this file included the customer ID, video name, frame, and camera number associated with detection, along with their projection coordinates on the 2D floor plan. The projections were calculated using the provided homography matrix from the dataset, ensuring that the field of view from each camera could be projected into a common floor plan [17].

### 2.3. Data Preparation

This section outlines the steps involved in the data preparation process applied to the dataset. Figure 4 provides a visual representation of the data preparation pipeline.

After obtaining the projections, we started data preparation by adjusting them to align with the floorplan's dimensions. This was achieved through a coordinate rescaling to fit the provided store map. Then, to obtain the exact time of each detection we used features such as video name and frame number, especially since each day's recording was segmented into shorter videos. The frame number corresponds to the detection frame within its corresponding video. Each video has a length (L) of 36 s, captured at 20 frames per second (fps), which corresponds to 120 of accelerated time. Therefore, the acceleration

factor (a) is determined as 720/120 = 6. The detection time is calculated based on the following formula:

$$\frac{(vn - 1) \times L \times fps + df}{a} = s \tag{1}$$

where $vn$ is the detection video number, $L$ is the video length, $fps$ is the video frame rate, $df$ is the frame number in the video where the detection was made, $a$ is the acceleration rate, and $s$ corresponds to the number of seconds from the time the camera started recording. After this, we just have to add the result to the time the camera started recording to find the time whose detection was identified. Consider an example where a person is present in video 4 at frame number 300, and the camera that captured this person started recording at 09:00:00. The time associated with the corresponding detection is calculated as follows:

$$\frac{3 \times 720 + 300}{6} = 410 \text{ s}$$

$$09:00:00 + 410 \text{ s} \longrightarrow 09:06:50$$

Given that we applied the tracking algorithm separately for each camera's videos, most of the customer IDs were duplicates. This duplication occurred because the IDs for each camera began with the number 1. To ensure the uniqueness of each ID, we adjusted the customer ID by adding the last ID of one camera to the IDs of the subsequent camera.

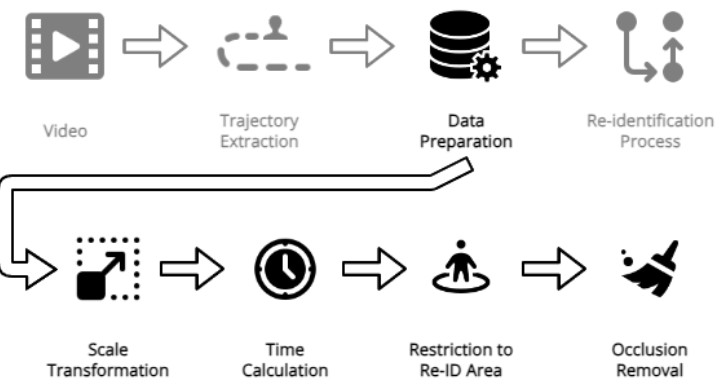

**Figure 4.** Data preparation pipeline.

After having the data projected in the store map, we noticed that the points too distant from the camera position were not accurate when compared with the trajectory performed by the customers. Such deviations could stem from errors in the homography matrix computation or from the inclusion of low-confidence detections. Given that these points were not relevant to our primary objective, we decided to focus on a specific area of interest (AoI) for the re-identification. The AoI is represented in Figure 5. The zones painted in black and gray correspond to shelves and map delimitation, respectively.

To enhance the quality of the available data, we used a filtering criterion by eliminating static points associated with a unique ID whose position did not change over time (i.e., removing customer data points that remain in a constant position across all related detections). Such a measure aims to mitigate errors from false positives that can be produced by the object detector. For example, a stationary object like a chair might be wrongly identified as a person. The position of the points belonging to that ID will not change over time, making it a good exclusion criterion. In contrast, real people are expected to move, especially when entering or exiting the camera's field of view. Therefore, this criterion will distinguish real customers from erroneous detections.

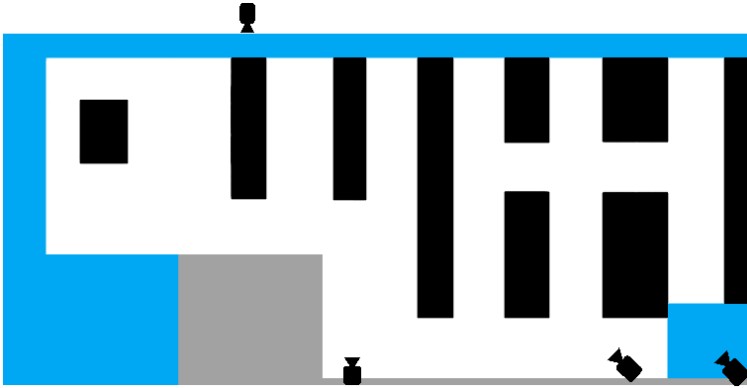

**Figure 5.** Map of the considered store area.

### 2.4. Re-Identification Process

The last step of the pipeline focuses on the re-identification process, which can be visualized in Figure 6. It is structured in two phases: initially, the goal is to re-identify points in close proximity, followed by finding a way to re-identify customers by making connections between the first and last points eligible for re-identification.

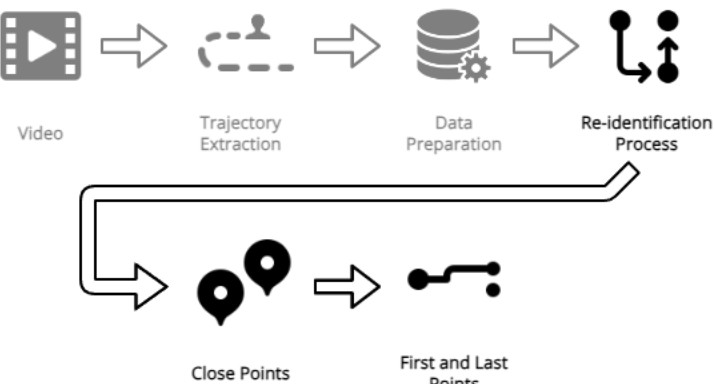

**Figure 6.** Re-identification pipeline.

The re-identification process starts by merging customer IDs whose points are close to each other but have a different ID due to problems in the detection and tracking phases. Such points typically originate from either a single camera or from two cameras that cover overlapping areas viewed from different perspectives. To execute this merging, a point-by-point iteration is conducted in chronological order. When there is a point with a low time gap close to the point we are iterating but with a different client ID, we establish that both points belong to the same person. The maximum permissible distance between these points must be less than the average distance between the trajectory points of two persons walking side by side. Once a point has been re-identified in this manner, it will no longer be considered for re-identification until this entire first step is completed.

In the second part of the process, we performed a distinction between two types of areas, the Exit/Entrance Areas and the Re-Identification Area, corresponding to the blue and white zones, respectively, as illustrated in Figure 2. The Exit/Entry Area represents an area where the points are not considered for re-identification since customers are likely to be leaving or entering the scene. The Re-Identification Area represents the area where points are allowed to be re-identified. The second part of the Re-Identification process starts by creating two groups. In one group, all the rows correspond to the customers' first positions, and in the other group, the rows are respective to the customers' last positions; both are ordered by time. The points' location must be present in the Re-Identification Area; otherwise, they are not considered for re-identification. Subsequently, we iterate over the group corresponding to the customers' last positions and, for every customer's

last position, we try to find, in the group of customers' first positions, one that is between a defined time interval from the time the last customer point was registered and inside a maximum distance range. If more than a point results from the applied criteria, we choose the customer to be re-identified based on the shortest distance to the last customer point. To calculate the distance from one point to another, we use the formula of Euclidean Distance between two points. We then take all the points that belong to the customer to be re-identified and associate a new ID that is equal to the customer ID of the last point and mark the re-identified customer so that they cannot be considered for re-identification again. If one of the last customer points does not have a first customer point that meets the required criteria, after the entire iteration process is complete, we assume that this point cannot be re-identified and the new customer ID remains the same.

Each customer ID is now associated with a new ID, which we called Customer Re-ID. However, as can be seen in the example of Table 1, the first four lines should all correspond to the same customer, i.e., customer 1. To ensure that a client is represented by only a single ID, we merged the pairs (Customer ID, Customer Re-ID) that had a common element in a list so that we could propagate the correct ID to all of them. After finishing the merge process, we changed the Customer Re-ID to the lowest value in the list. Using Table 1 as an example, we can see that after applying this process, the Customer ID in the first four lines will be 1 and the fifth line will remain the same.

**Table 1.** Re-identification result example.

| Customer ID | Customer Re-ID (Before Merging) | Customer Re-ID (After Merging) |
|:---:|:---:|:---:|
| 1 | 1 | 1 |
| 2 | 1 | 1 |
| 3 | 2 | 1 |
| 4 | 3 | 1 |
| 5 | 5 | 5 |

## 3. Results

In this section, we provide a description of the test results, their interpretation, and the conclusions that can be drawn. Firstly, we present the experimental setup used to carry out the tests, then we present the conducted tests and associated results, and finally we present the interpretation of these results and the limitations that were solved along the development of our framework.

### 3.1. Experimental Setup

Regarding the ByteTrack parameters, several experiments were conducted with different configurations, and the values that yielded the best results are presented in Table 2.

**Table 2.** ByteTrack parameters.

| track_buffer | fps |
|:---:|:---:|
| match_thresh | 0.8 |
| track_thresh | 0.5 |
| aspect_ratio_thresh | 1.6 |
| min_box_area | 10 |

It is also important to mention that we tried changing the position and velocity weights of the Kalman Filter, but these modifications did not produce consistent improvements, so the original values were kept.

The ByteTrack algorithm implementation is available in the author's github repository [11].

### 3.2. Experimental Process

The Re-Identification framework was developed with the purpose of re-identifying customers when they lose their ID either due to occlusion within the same camera view or to changing their presence from one camera view to another. Since this dataset contains no customer ground-truth positions, we opted to annotate a set of examples, registering the number of persons present in each example to evaluate the current approach. This procedure allowed for the comparison of the number of generated IDs and Re-IDs with the number of actual persons present in the store. It also detailed the number of re-identifications made. Table 3 provides an overview of each of the annotated examples.

**Table 3.** Re-identification results.

| Example Number | N° Persons in Scene | Total N° Re-IDs | N° Wrong Re-IDs | N° Customer IDs | N° Customer Re-IDs |
|---|---|---|---|---|---|
| 1 | 2 | 16 | 1 | 20 | 2 |
| 2 | 2 | 5 | 1 | 17 | 3 |
| 3 | 2 | 8 | 1 | 10 | 3 |
| 4 | 1 | 13 | 3 | 19 | 4 |
| 5 | 5 | 15 | 2 | 33 | 7 |
| 6 | 2 | 4 | 0 | 15 | 2 |
| 7 | 3 | 3 | 1 | 13 | 4 |
| 8 | 3 | 12 | 2 | 18 | 5 |
| 9 | 4 | 15 | 3 | 30 | 7 |
| 10 | 6 | 11 | 1 | 20 | 7 |
| 11 | 3 | 2 | 2 | 11 | 5 |
| 12 | 6 | 7 | 0 | 29 | 6 |
| 13 | 2 | 15 | 2 | 27 | 4 |
| 14 | 2 | 6 | 0 | 13 | 2 |
| 15 | 2 | 4 | 0 | 11 | 2 |
| 16 | 2 | 8 | 0 | 14 | 2 |
| 17 | 1 | 2 | 1 | 7 | 2 |
| 18 | 3 | 6 | 8 | 38 | 11 |
| 19 | 3 | 7 | 0 | 24 | 3 |
| 20 | 3 | 13 | 1 | 26 | 4 |
| 21 | 5 | 13 | 3 | 24 | 8 |
| 22 | 6 | 20 | 5 | 31 | 5 |
| 23 | 1 | 5 | 1 | 10 | 2 |
| 24 | 3 | 8 | 1 | 20 | 4 |
| 25 | 4 | 7 | 1 | 32 | 5 |
| Total | 76 | 225 | 40 | 512 | 109 |

The results indicate that the number of identifiers without re-identification applied is 512. After the re-identification process, the number of total IDs was reduced to 109, which is much lower than the number of original IDs. However, it is still higher than the number of customers actually present in the considered area of the store (N = 76).

To have a better understanding of the results, we made a comparison between a version with the trajectories painted according to the IDs provided by the object tracker and a version with the trajectories painted according to the IDs generated after applying the re-identification process (Re-IDs). Figures 7 and 8 illustrate this comparison.

Comparing Figures 7 and 8, we can see that there are two people in the recordings, one standing still (dots in the bottom left corner of the images) and the other moving. The trajectory of the moving person was identified by the tracker with 11 different IDs, and our algorithm was able to join these trajectory fragments and associate them with a single customer identifier. There are also some pink dots in Figure 7, which are the result of a detection error in the output of the object detector, which are removed by applying pre-processing operations.

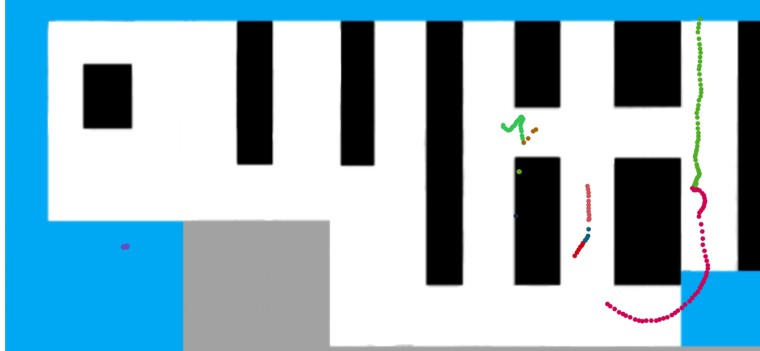

**Figure 7.** Example with tracker IDs projected.

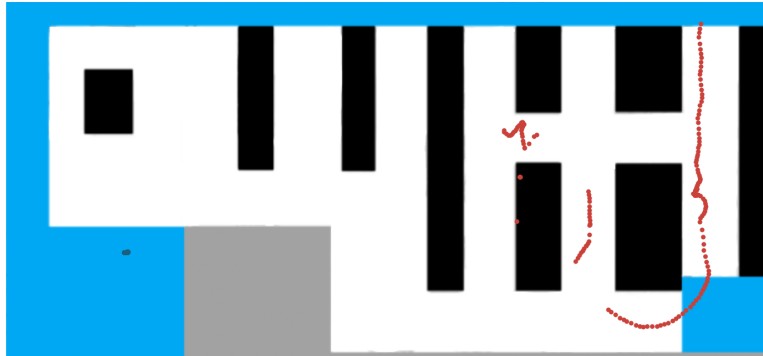

**Figure 8.** Example with Re-IDs projected.

Occlusions represent a constraint in the trajectory re-identification quality because, in cases where customers cannot be detected because they are occluded by an object (usually shelves), their trajectory cannot be extracted. This results in gaps in the trajectory that cause difficulties for the algorithm to correctly associate and re-identify customers.

In Figure 9, the three sets of trajectory points, highlighted with a red circle, should be re-identified as two customers, but as there is no information about the trajectory of one of the customers when they are behind a shelf, the algorithm is not able to associate the trajectories shown to two different customers, thus presenting a limitation in the re-identification process.

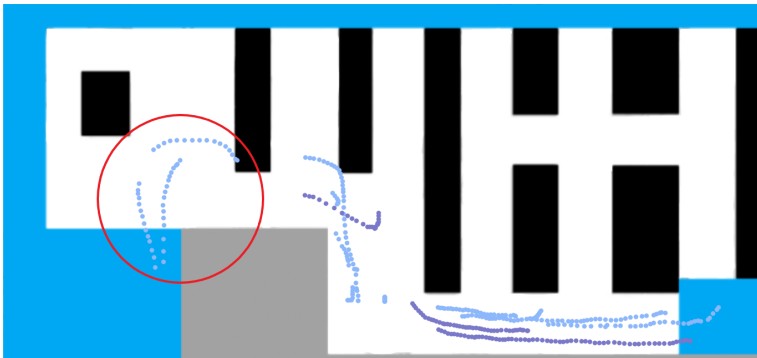

**Figure 9.** Trajectory failure making it impossible to correctly re-identify.

That said, we decided to calculate the re-identification success rate. The formula below shows how the re-identification success rate was obtained:

$$ReIDSuccessRate = 1 - \frac{\text{N}^\circ \text{ Wrong ReIDs}}{\text{Total N}^\circ \text{ ReIDs}}$$

The number of wrong re-identifications includes over- and under-re-identifications, i.e., re-identifications that were not necessary because they combined two different clients or re-identifications that were necessary but were not carried out. The total number of re-identifications for all the examples adds up to a total of 225, and the total number of wrong re-identifications was 40. As a result, the hit rate is equal to 0.82(2), i.e., the re-identification algorithm developed was able to re-identify approximately 82 percent of the customers identified in the considered examples. It is important to note that this value does not take into account the reduction in identifiers caused by the pre-processing applied to the dataset, i.e., the value presented only refers to the re-identification process.

### 3.3. Framework Improvements and Limitations

During the development of the algorithm, several problems were successfully solved, or their effects mitigated. These are detailed in the following subsections.

### 3.3.1. Overlapping Areas

The first part of our algorithm, mentioned in Section 2.4, was implemented to solve a problem where points present in areas covered by more than one camera have different IDs, when in fact they correspond to the same person but were extracted from different cameras. This would introduce extra IDs when there were data from more than one camera in a single region of the store.

### 3.3.2. Exit/Entrance (E/E) Areas

In certain cases, such as a customer losing their ID in an E/E Area (areas marked in blue), the re-identification process cannot be performed. This loss of ID is due to the fact that the first body part to appear is usually the customer's head, which is blurred (for anonymization purposes). This leads to a scenario where the tracker is unable to identify the person in the next frame due to its inability to associate the blurred image from the previous frame with the same individual when the entire body is visible. This problem was also solved with the first part of the re-identification process. Figure 10 shows an example where a few points, present in the bottom left and bottom right E/E Areas, had different IDs (marked with different colors) due to the previously described reason.

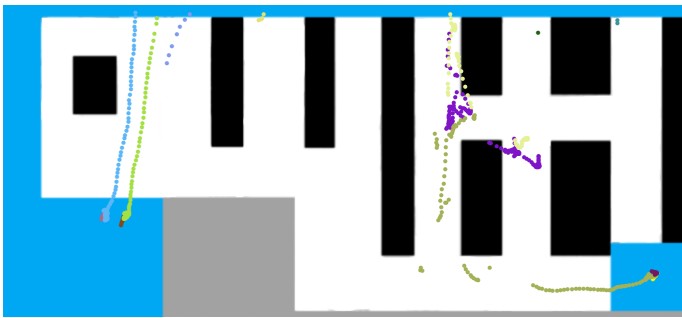

**Figure 10.** E/E Area solved problem.

### 3.3.3. Occlusion Translation

Another issue, also related to the environmental complexity due to occlusions, is the deformation of part of the trajectories caused by the presence of shelves. This mostly happened when the data were extracted from a specific camera (bottom right corner) and when the customer is behind one or more shelves. The result was a coordinated translation and, in some cases, it affected the re-identification process, in the way that the trajectory part that suffered the translation can be out of the allowed distance to be re-identified. To mitigate this issue, when the customer is present in areas covered by more than one camera, we only use the data from cameras that are not affected by occlusions.

### 3.3.4. Limitations

Regarding the framework limitations, as we previously described, occlusions caused by objects such as shelves or counters are one of the main obstacles for the re-identification tool created. Since it works with distances and time intervals between trajectory points, its operation is affected when there are large losses in what would have been the original route taken by customers. Other limitations occur when some factor (occlusion or distance from camera) makes it impossible to identify a customer present in an E/E Area for a long period of time. When the customer reappears, the algorithm will not be able to make an association with its previous identification because the customer is present in an E/E Area and, when present in that area, the time for re-identification is short.

### 4. Conclusions

In this study, we developed a system for detecting and tracking customers in a commercial retail environment using videos captured with video surveillance cameras in a multi-camera environment. These videos are blurred in the area of customers' faces in order to keep their anonymization, making it difficult to use traditional image-based re-identification methods. Thus, this study proposes a mechanism for re-identifying people based on the paths they have traveled in the store. The trajectories are obtained based on information from the detection and tracking system which, after detecting and associating an identifier to each person, calculates and extracts the trajectory taken by them.

The tests conducted show the effectiveness of the proposed trajectory-based re-identification mechanism in associating a unique identifier to each customer regardless of the camera they were detected in. Our method was capable of successfully re-identifying 82% of the customers present in the considered examples. This approach can be seen as a way to improve the accuracy of tracking algorithms, allowing them to minimize the number of identifiers generated, and presents an alternative to the use of image-based person re-identification processes in a commercial environment with multiple cameras. It is also a solution for dealing with some occlusions and errors caused by object detectors and tracking algorithms, a solution to consider when the data we have have part of the person occluded due to privacy issues or when we simply only have access to the trajectory data.

As for future work, optimizing the re-identification process will require rigorous testing across varied datasets. Central to this line of work is the understanding that the accuracy of current detection and tracking processes lies in the precision of the algorithms employed. Therefore, exploring alternatives or enhancing the existing ones will be important to achieve superior results. Moreover, the context in which the cameras operate is very relevant. We anticipate that in environments with fewer occlusion challenges than our current dataset, the re-identification mechanism would obtain better results. Therefore, we expect that the strategies we provided in this work by incorporating occlusion-aware mechanisms will improve the trajectory quality by mitigating the errors arising from occlusions, as also discussed in [18]. Another suggestion is to add confidence scores to re-identifications, since they can offer a more nuanced perspective on their reliability. Finally, it is worth noting that our framework, in its current form, was not developed for real-time scenarios. However, with suitable modifications, transitioning it to real time is a feasible step.

In conclusion, this work utilizes images obtained in a complex retail environment, with constraints such as occlusions and data privacy concerns, to re-identify customers in large retail stores. It contributes with a different approach to the topic of person re-identification in multi-camera environments since it only uses trajectory data to accomplish this re-identification and represents a possible alternative to the traditional methods based on images.

**Author Contributions:** Conceptualization, T.B., L.N., P.A. and P.J.; methodology, T.B., L.N. and P.A.; software, D.M. and S.C.; validation, D.M.; investigation, D.M., P.J. and S.C.; resources, L.N.; data curation, L.N. and D.M.; writing—original draft preparation, D.M.; writing—review and editing, L.N., T.B., P.A. and S.C.; visualization, D.M.; supervision, L.N., T.B. and P.A.; project administration, L.N. All authors have read and agreed to the published version of the manuscript.

**Funding:** This work was partially supported by Fundação para a Ciência e a Tecnologia, I.P. (FCT) [ISTAR Projects: UIDB/04466/2020 and UIDP/04466/2020] and P2020 project "ECI4.0": LISBOA-01-0247-FEDER-047155.

**Institutional Review Board Statement:** Not applicable.

**Informed Consent Statement:** Customer consent was waived because only blurred images were processed and no identifiable images were published.

**Data Availability Statement:** Used data were made available under the "Espaços Comerciais Inteligentes 4.0" (ECI4.0) project and they may not be published to respect the privacy agreement.

**Conflicts of Interest:** The authors declare no conflict of interest.

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
