# Peer review of "Multi-Camera Person Re-Identification Based on Trajectory Data"

_applsci, doi:10.3390/app132011578_

Round 1
Reviewer 1 Report
1. The framework of the whole method is missing. A new figure should be added to show the pipeline.
2. For the experimental section, the experimental comparison with different methods should be reported.
3. Sec 2.1. details Dataset Description. The dataset is small, especially compared to the existing reid datasets
4. The reference section is short. Some more recent related works[1,2,3,4,5] should be cited for more comprehensive literature survey. [1] Prompt Learns Prompt: Exploring Knowledge-Aware Generative Prompt Collaboration For Video Captioning,IJCAI 2023 [2] E^2VPT: An Effective and Efficient Approach for Visual Prompt Tuning, neurips [3] Coarse-to-fine video instance segmentation with factorized conditional appearance flows,IEEE/CAA Journal of Automatica Sinica [4] Learning hierarchical embedding for video instance segmentation,Proceedings of the 29th ACM International Conference on Multimedia [5] Sg-net: Spatial granularity network for one-stage video instance segmentation,Proceedings of the IEEE/CVF Conference on Computer Vision and Pattern
The overall qulity of the writing is good. There are some tiny issues, for instance, Store map considered area. It is not clear.
Reviewer 2 Report
This study introduces a trajectory-based person re-identification tool for tracking customers in a multi-camera retail environment, addressing challenges like occlusions and privacy concerns. It achieved an 82% re-identification rate, showcasing its effectiveness in real-world scenarios.
However, there are some limitations are highlighted as follow:
1- Some grammatical errors are there, it suggested to review the paper language carefully.
2- Abbreviations such as CNN should be defined inside the text once it appears.
3- It is advised to add a graph describing the system flowchart.
4- adding measurement criteria validation is essential for a robust evaluation of the proposed method. To enhance the credibility of the study, the authors should consider incorporating the following measurement criteria:
a. Re-identification Accuracy: Quantify the accuracy of person re-identification by measuring the percentage of correctly matched trajectories. This accuracy metric should consider both true positives (correctly matched trajectories) and false positives (incorrect matches).
b. Tracking Consistency: Assess the tracking consistency over time by measuring metrics such as track fragmentation and track merging.
c. Robustness to Occlusions: Evaluate how well the method handles occlusions, both partial and full, by measuring the rate of successful re-identifications in occluded scenarios.
d. Computational Efficiency: Measure the processing time required for tracking and re-identification to determine if the tool is suitable for real-time applications in a large retail store environment.
Seems good. a few modifications are needed.
Round 2
Reviewer 2 Report
The table Caption should be upper the table, revise Table 1,2,3.
Author Response
Dear Reviewer,
Thank you for accepting our work for publication. The captions were already fixed and a final version with the applied changes will be submited.
Once again, thank you for your work and feedback, that contributed to improve the quality of our work.
Best reegards,
The Authors